# CART-RF Classification with Multifilter for Monitoring Land Use Changes Based on MODIS Time-Series Data: A Case Study from Jiangsu Province, China

**Le'an Qu [1,2], Zhenjie Chen [1,3,*] 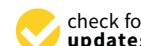 and Manchun Li [1,3]**

[1] School of Geography and Ocean Science, Nanjing University, Nanjing 210023, China;
qulean@ahnu.edu.cn (L.Q.); limanchun@nju.edu.cn (M.L.)
[2] School of Geography and Tourism, Anhui Normal University, Wuhu 241002, China
[3] Jiangsu Provincial Key Laboratory of Geographic Information Science and Technology, Nanjing University, Nanjing 210023, China
* Correspondence: chenzj@nju.edu.cn; Tel./Fax: +86-25-8968-1185

**Abstract:** The periodic determination of land use changes over large areas is crucial for improving our understanding of land system dynamics. Jiangsu lies at the center of China's Yangtze Delta and has one of the fastest-developing economies in China. However, it is also a region where serious conflicts exist between the available land resources and the human demand for land. To address these conflicts, it is important to analyze the patterns of land use change in Jiangsu, as they can serve as a useful reference for other rapidly urbanizing regions in China as well as other developing countries. In this study, we propose a method of classification and regression tree-random forest (CART-RF) classification with a multifilter based on time-series Moderate Resolution Imaging Spectroradiometer (MODIS) imaging data. The proposed method integrates the CART decision tree and the random forest algorithms (CART-RF) to obtain accurate yearly land use data for large areas from multivariate time-series remote sensing data and employs a spatial-temporal-logical filter to exclude any abnormal changes in the multivariate time-series pixel data. The obtained results indicated that (1) the CART-RF classifier is effective for land use classification based on the multivariate time-series MODIS data, with the overall classification accuracy being greater than 90%; (2) the use of the proposed combinatorial spatial-temporal-logical filtering method effectively eliminates most anomalous changes and minimizes the effects of "salt-and-pepper" noise; and (3) from 2000 to 2015, land use in Jiangsu province underwent significant and spatiotemporally heterogeneous changes on a province-wide scale, owing to various factors, such as those related to the economy, location, and government policies. These changes were manifested as continuous expansions in the built-up land at the expense of farmland. While this expansion of built-up land has been very rapid in southern Jiangsu, especially in the region close to Yangtze River Delta, it has been relatively slower in northern Jiangsu.

**Keywords:** land use change; CART-RF classification; time-series data; spatial-temporal-logical filter; Jiangsu

## 1. Introduction

In the last few decades, the rapid development of human society has been accompanied by rapid urbanization on a global scale. Consequently, human activity has had profound effects on land use and land cover all over the world [1,2]. Land use change has severely altered the Earth's ecosystems, and these changes will continue to have an adverse effect on the sustainable development

of the world [3,4]. Jiangsu lies at the center of China's Yangtze River Delta (YRD) and has one of the fastest-developing economies in China. Since the beginning of the 21st century, the gross domestic product (GDP) of Jiangsu province has increased from 857.4 billion CNY in 2000 to 7011.6 billion CNY in 2015. Further, the urbanization and industrialization of this region show no signs of pausing. Meanwhile, Jiangsu province is also an important grain production region in China because of its wet weather, abundant sunshine, flat terrain, and fertile soil. The rapid development of Jiangsu province has led to extensive urban expansion and significant farmland losses. Conflicts between the need for development and the necessity of protecting farmland have become increasingly prominent over time. In addition, there are significant imbalances in the internal economic development of Jiangsu province. Although southern Jiangsu is one of the most rapidly developing regions in the world, the economic development of northern Jiangsu has been relatively slower [5]. Therefore, analyzing the land use change patterns in Jiangsu province is of great importance, as these patterns can serve as a useful reference for the formulation of land use policies and land use planning in other rapidly urbanizing regions.

Remote sensing technology allows for a dynamic, fast, accurate, and comprehensive overview of land use over wide areas and is now being used widely in the monitoring of land use changes [6–9]. Rapid developments in Earth observation technologies have led to an abundance of remote sensing datasets. Most current studies on the remote-sensing-based monitoring of land use changes have focused on long-term land use changes. In these studies, the regional land use changes are usually monitored by classifying and comparing single images of a representative time-phase during each year (e.g., the time when summer vegetation flourishes) [10–12]. However, these methods are strongly affected by the imaging conditions as well as by seasonal ground object variations. Therefore, researchers are increasingly using time-series remote sensing (TSRS) data for the detection of land use changes [13,14]. As TSRS data contain both spectral and temporal information, they provide abundant information regarding the vegetation phenology and help minimize ambiguities in feature identification, thus improving the accuracy of land use classification [15].

The remote sensing images acquired by the Moderate Resolution Imaging Spectroradiometer (MODIS) satellite cover a large area in each image and have high temporal resolution. This makes MODIS data highly advantageous for the monitoring of long-term land use changes over large areas [16,17]. In recent years, there have been many studies on land use changes using time-series MODIS images, including those involving the determination of agricultural regions [18,19] and woodland regions [20,21], the classification of land use [22–24], and the determination of land use changes [25,26]. However, these studies used univariate time-series data to extract the land use change information, making them difficult to classify land use in a comprehensive and effective manner. In reality, the classification accuracy associated with different remote sensing indexes will change significantly with the same method, owing to the complexity of the actual ground surfaces [27,28]. Furthermore, since the spatial resolution of MODIS images is low, mixed pixels occur frequently, leading to "salt-and-pepper" noise in the land use classification results. For land use classification, "salt-and-pepper" noise refers to the random misclassified pixels surrounded by other land use classes, for instance, several farmlands in the center of a lake [29]. "Salt-and-pepper" noise reduces the accuracy of land use change detection [30]. Therefore, it is necessary to use multivariate time-series MODIS data to develop a robust classification method, in order to improve the accuracy of land use change detection over long time scales and large areas.

In this study, we propose a method of classification and regression tree-random forest (CART-RF) classification with a multifilter based on multivariate TSRS imaging data from MODIS. The method uses combinatorial spatial-temporal-logical filtering to remove most pixels with anomalous changes. We used the method to obtain land use in Jiangsu province from 2000 to 2015. We also analyzed the spatiotemporal processes associated with the land use changes in Jiangsu province. The objective of this study was to answer the following questions: (1) how can the anomalous spatiotemporal changes in individual pixels be eliminated to allow for the classification of yearly land use over large areas with

high accuracy?; (2) what are the characteristics of land use changes in Jiangsu province?; and (3) how do land use changes in this province vary during each period and in different regions?

## 2. Study Area and Data Sources

### 2.1. Study Area

Jiangsu is located at the center of the eastern coast of mainland China. Its geographical coordinates are 116°18′–121°57′E, 30°45′–35°20′N. Jiangsu province consists of 13 cities, and can be divided into three major regions in terms of economy and location: southern Jiangsu, northern Jiangsu, and central Jiangsu. Southern Jiangsu consists of Nanjing, Zhenjiang, Changzhou, Wuxi, and Suzhou. Central Jiangsu consists of Yangzhou, Nantong, and Taizhou. Northern Jiangsu includes Xuzhou, Lianyungang, Suqian, Huai'an, and Yancheng (Figure 1). Jiangsu has a transitional climate that varies between temperate and subtropical climes. Southern Jiangsu has a wet and warm subtropical monsoon climate, whereas north Jiangsu has a cold continental monsoon climate. The area of Jiangsu province is 107,200 km², with farmland and built-up land accounting for more than 50% and 20% of its land, respectively.

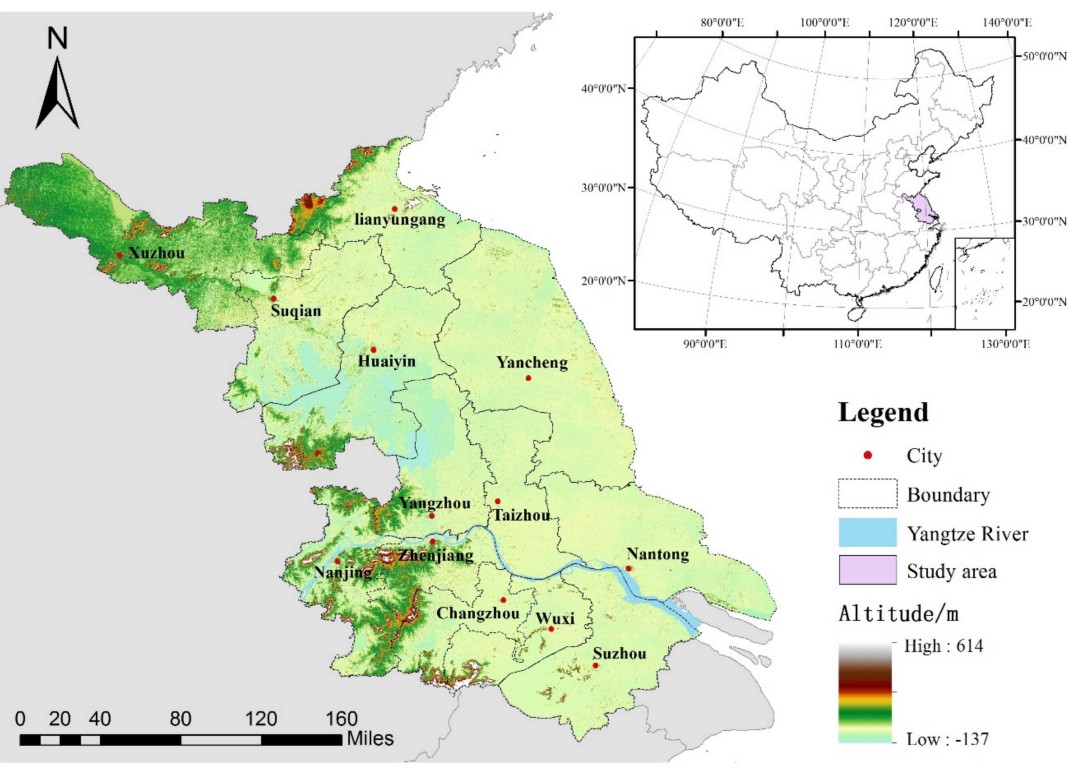

**Figure 1.** Location of study area.

Jiangsu is one of the most economically developed provinces in China, and its population and GDP are both increasing rapidly. In 2015, Jiangsu accounted for 5.8% of China's population. Its GDP accounts for 10.4% of China's, ranking second in China. Its per capita GDP is 1.8 times more than the average of China, ranking the fourth in China. Based on the Jiangsu Statistical Yearbook [31], the populations and GDPs of southern Jiangsu, central Jiangsu, and northern Jiangsu increased significantly from 2000 to 2015 (see Table 1). This growth in the population and economic development have been accompanied by the rapid urbanization of Jiangsu, as well as great changes in its land use, namely, the loss of farmland and the expansion of urban areas.

**Table 1.** Total population, urban population, and gross domestic product (GDP) of study area.

| Region | Population in 2000 (Million) | Population in 2015 (Million) | Urban Population in 2000 (%) | Urban Population in 2015 (%) | GDP in 2000 (Billion CNY) | GDP in 2015 (Billion CNY) |
|---|---|---|---|---|---|---|
| Southern Jiangsu | 21.7 | 33.2 | 59.6 | 75.3 | 481.5 | 4151.9 |
| Central Jiangsu | 17.4 | 16.4 | 37.7 | 62.4 | 161.4 | 1385.3 |
| Northern Jiangsu | 31.6 | 30.1 | 31.2 | 59.1 | 197.6 | 1656.4 |

*2.2. Data Sources*

The data sources used in this study included the following: (1) 2000–2015 MOD13A1 Normalized Difference Vegetation Index (NDVI) dataset products. NDVI data have a temporal resolution of 16 days and spatial resolution of 500 m. A total of 365 images were collected; (2) 2000–2015 MOD09A1 dataset products, with a temporal resolution of 8 days and spatial resolution of 500 m. A total of 728 images were collected. The MOD09A1 dataset was mainly used to calculate the Normalized Difference Water Index (NDWI) and Normalized Difference Soil brightness Index (NDSI); (3) land survey data from 2000, 2005, and 2010 with a scale of 1:10,000. The land survey data were used to assess the accuracy of the CART-RF classifications. The land survey data includes five types of land use (farmland, woodland, grassland, built-up land, and water body); (4) administrative boundary data for Jiangsu province; (5) digital elevation model data for Jiangsu province; and (6) population and socioeconomic data for Jiangsu province. The details of these data are listed in Table 2.

**Table 2.** Details of data used in this study.

| Data Source | Data Acquisition Period | Resolution/Scale | Data Format | Source | Usage |
|---|---|---|---|---|---|
| MODIS Images (MOD13A1) | 2000–2015 | 500 m | GeoTiff | NASA (USA) | Extraction of NDVI time-series |
| MODIS Images (MOD09A1) | 2000–2015 | 500 m | GeoTiff | NASA (USA) | Calculation of NDWI and NDSI time-series |
| Land Survey Data | 2000, 2005, 2010 | 1:10,000 | ShapeFile | Provincial Geomatics Center of Jiangsu | Validation of land use classification accuracy |
| Administrative boundary data | 2015 | 1:10,000 | ShapeFile | Provincial Geomatics Center of Jiangsu | Determination of boundaries of each city in Jiangsu |
| Digital Elevation Model Data | 2009 | 30 m | GeoTiff | Geospatial Data Cloud | Elevation data |
| Population and socioeconomic data | 2000–2015 | / | / | Jiangsu Statistical Yearbook | Analyzing factors influencing regional land use changes |

## 3. Method

To improve the accuracy of land use change detection over large areas, we developed a method of CART-RF classification with a multifilter based on the multivariate TSRS data. In this method, the CART-RF classification model was employed to improve the distinguishability of each type of land use, while combinatorial spatial-temporal-logical filtering was used to minimize the effects of

"salt-and-pepper" noise by eliminating the pixels with anomalous changes (Figure 2). The proposed method involves the following four steps: (1) TSRS construction: the NDVI, NDWI, and NDSI values were calculated from each time each scene of MODIS images and were used to construct the time-series for each pixel; (2) stable pixel detection: the stable pixels, that is, those whose corresponding land use did not change over the period of study, were extracted to act as reliable references for land use classification; (3) CART-RF classification: the CART-RF classification model was used to perform preliminary land use classification; and (4) combinatorial filtering: spatial-temporal-logical filtering was performed on the preliminary land use classification results to eliminate most anomalous pixels and improve the accuracy of classification. These steps are shown in Figure 2.

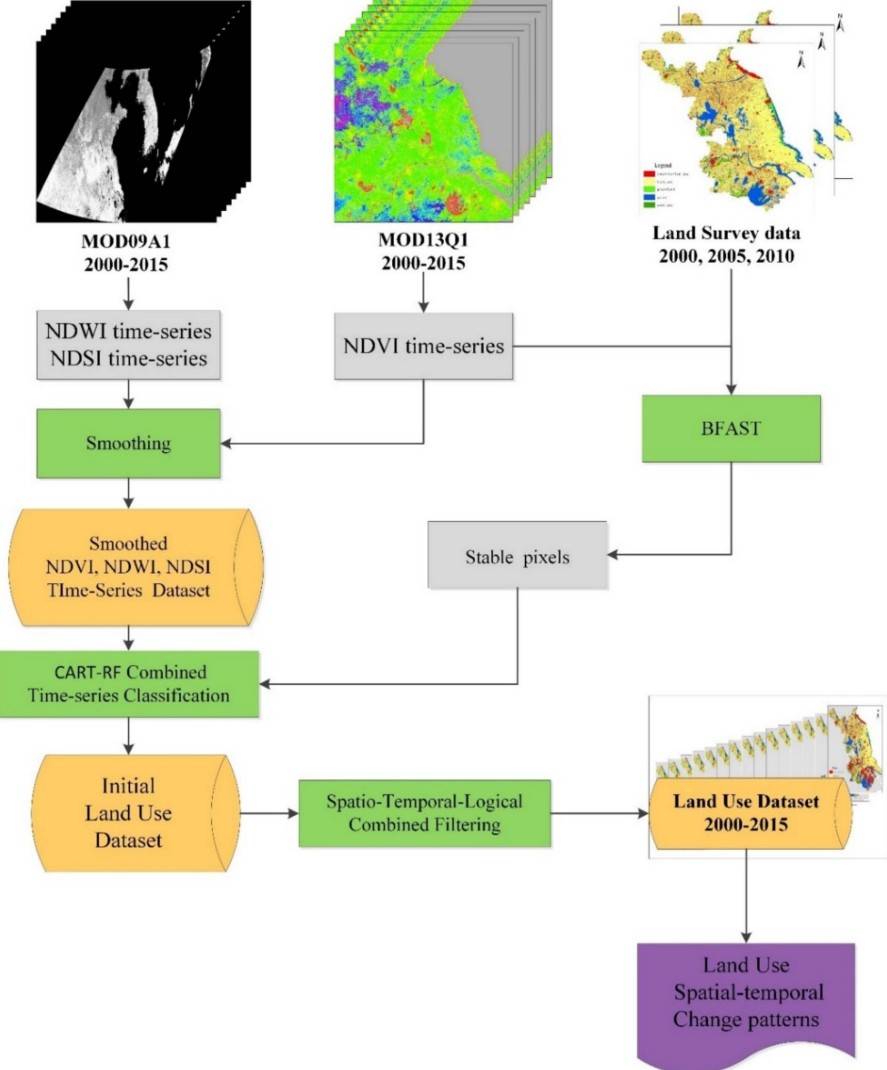

**Figure 2.** Flowchart of proposed approach for land use classification. NDVI, NDWI, and NDSI are Normalized Difference Vegetation Index, Normalized Difference Water Index, and Normalized Difference Soil brightness Index, respectively.

### 3.1. Construction of Multivariate Remote Sensing Image Time-Series

Given the complexity of land use patterns, it is difficult to effectively classify land use based on a single index. Hence, in this study, NDVI, NDWI, and NDSI were used to construct a multivariate time-series, which was then used for land use classification. The NDVI is an effective indicator of seasonal changes in high-vegetation-cover areas, as it is highly sensitive to differences in vegetation [32]. The NDWI can be used for highlighting the differences between water bodies and other types of land

use [33]. The NDSI is based on the differences between the soil background reflectance of the various types of land cover during each season; this index may, therefore, be used to extract the vegetation features for land use classification [34]. To reduce the computational workload, the NDVI time-series was constructed by extracting the NDVI data from the MOD13A1 dataset using the MODIS Reprojection Tool and then combining this dataset with the NDVI dataset calculated from the MOD09A1 dataset. The NDWI and NDSI time-series were calculated from the MOD09A1 dataset using the software ENVI. Finally, the spatial and temporal resolutions of the multivariate time-series were set to be uniform at 500 × 500 m and 8 days, respectively. The equations for calculating NDVI, NDWI, and NDSI are as follows:

$$NDVI = \frac{ref2 - ref1}{ref2 + ref1}, \tag{1}$$

$$NDWI = \frac{ref4 - ref2}{ref4 + ref2}, \tag{2}$$

$$NDSI = \frac{ref1 - ref4}{ref1 + ref4}, \tag{3}$$

where *ref* 1, *ref* 2, and *ref* 4 are the reflectance of Band 1, Band 2, and Band 4, respectively, of the MOD09A1 dataset.

　　As the MODIS time-series data are affected by various interference-related factors (i.e., the cloud cover, rainfall, and solar azimuth angle, among others), the curves for multivariate time-series extracted from MODIS datasets often exhibit irregular, sawtooth-shaped fluctuations, making the task of land use classification a difficult one. Therefore, it is necessary to remove the fluctuations or noise from these time-series. Thus, smooth and even curves were reconstructed for the time-series [35]. The multivariate time-series was reconstructed using an asymmetric Gaussian function. The reconstructed time-series described the long-term trends and seasonal changes in the land use with greater fidelity [36]. The software TIMESAT 3.3 was used to smoothen and reconstruct the multivariate time-series data. The smoothed NDVI curves are shown in Figure 3.

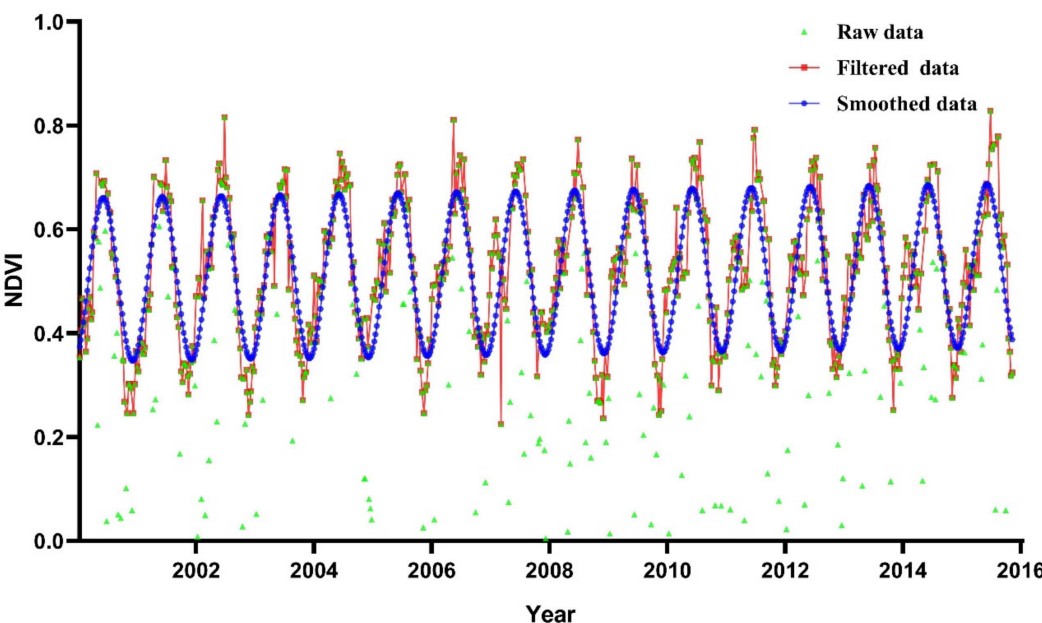

**Figure 3.** Time-series filtering and smoothing of reference data sample.

## 3.2. Detection of Stable Pixels

　　During TSRS-imaging-data-based land use classification, it is necessary to select a few stable pixels that can be considered reliable references for the classification of the different land use types. The land

use types of these stable pixels must remain unchanged throughout the study period. A reference time-series for each type of land use can be obtained from the time-series of stable pixels [37]. To filter the seasonal changes in the time-series, the Breaks for Additive Season and Trend (BFAST) method was used to decompose the NDVI time-series into long-term trends, seasonal phenology, and residuals through iterative decomposition. This method can be used to extract the time and magnitude of the NDVI changes and is highly robust [38]. By analyzing the long-term trends in the NDVI time-series, stable pixels with small NDVI fluctuations can be identified.

The procedure of BFAST-based identification of the stable pixels is as follows. First, the BFAST method is used to determine the long-term trends from the NDVI time-series. The magnitude of the changes in the NDVI corresponding to these long-term trends is analyzed for each pixel, and the pixels that only exhibit small fluctuations in the NDVI over the period of interest are extracted. Since no land use changes would have occurred in most parts of the area of study, most of the pixels in the remote sensing images would be extracted by this process. Then, median filtering with a $3 \times 3$ majority filter is performed to remove the erroneous pixels from the marginal areas; this is particularly important for landscapes with small and heterogeneous land cover entities. Only relatively stable areas are retained. In this manner, the noisy pixels, such as those corresponding to small or marginal areas, are removed. Finally, the stable areas are subjected to unsupervised classification (into five land use types) to produce spectrally homogeneous regions. Only the central pixels that are also noise free are extracted and regarded as stable pixels in order to reduce the edge effects.

To ensure that the sample pixels are representative of all the land use types in Jiangsu, stratified sampling was performed to select stable pixels for each type of land use. Further, it was ensured that the sampled pixels were distributed over least 1000 points per land use type to obtain an unbiased distribution of samples across the study area. The land survey data were used to crosscheck the sample pixels and identify the land use types of these pixels. The smallest distance during stratified sampling was set to 1500 m in order to prevent spatial autocorrelation between the samples [39]. We extracted sample pixels that covered an area corresponding to at least $3 \times 3$ contiguous pixel units.

### 3.3. CART-RF Classification of Multivariate Time-Series

The CART model is a decision tree model that is used frequently for classifying remote sensing images because it is simple, fast, yields good results, and is capable of handling large sets of high-dimensionality data [40]. However, it is difficult to accurately distinguish between different types of land use using the CART classifier alone. The random forest (RF) method is an ensemble learning method that can be used to combine multiple classification models and increase the overall classification accuracy. The RF classifier uses bootstrap aggregation and the random subspace method to construct a large number of decision trees; classification is then performed via majority voting [41]. The CART-RF method combines CART with the RF model. This method has the ability to classify complex features and is highly robust against noisy data as well as data with missing values. However, if the CART-RF model is used with a small number of time-phases, it is unable to identify the phenological features of the ground objects with the desired accuracy. Hence, seasonal changes may significantly affect the accuracy of land use change detection. Therefore, land use classification was performed by using the CART-RF algorithm on multivariate time-series data (Figure 4).

Two parameters must be defined when using the CART-RF model, namely, the number of variables in each node (Mtry) and the number of trees (Ntree). It is generally thought that Mtry should not exceed the square root of the total number of input variables and that the errors of RF models usually stabilize before Ntree increases to 500 [42]. Based on repeated cross-validation tests, we selected Mtry = 1 and Ntree = 100 as the optimal parameters for the CART-RF model. First, the RF model was trained using the NDVI, NDWI, NDSI, and spectral data of the samples. The RF model was then used to create multiple RF voting matrices. The results of the NDVI and spectral voting processes were then used by the CART model to determine the high- and low-vegetation-cover areas. Based on the results of NDSI voting, the high-vegetation-cover areas were either classified as farmland, woodland, or grassland by

the CART model. The low-vegetation-cover areas were classified either as water bodies or built-up land by the CART model, based on the results of NDWI voting. Finally, all the classified land use types (i.e., woodlands, farmlands, grasslands, built-up lands, and water bodies) were combined to obtain the land use classification map.

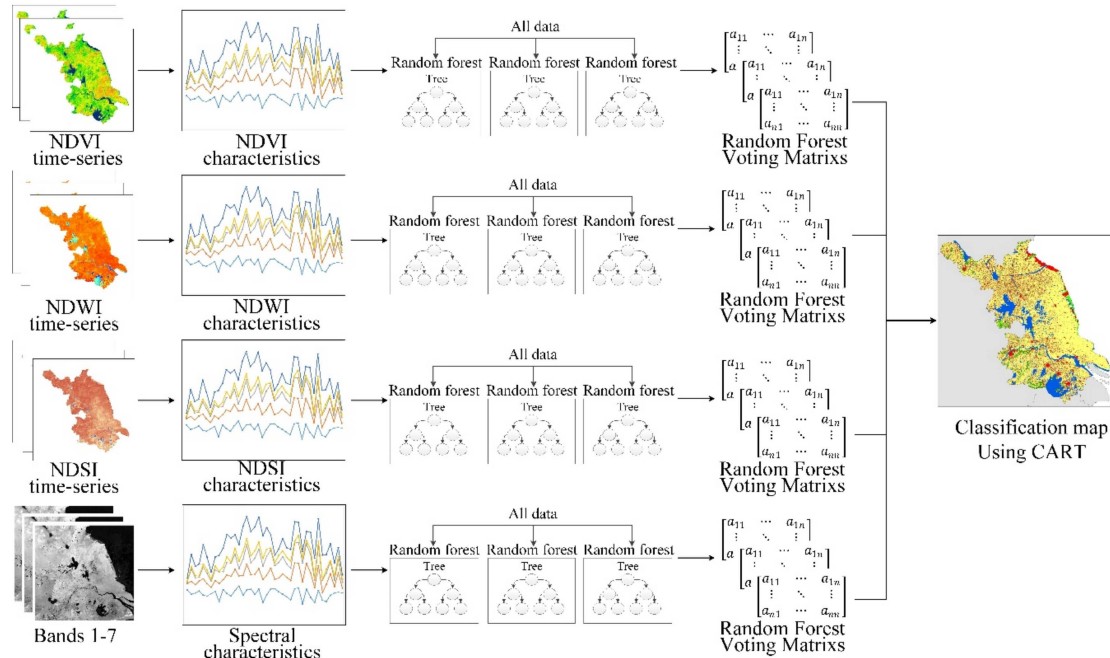

**Figure 4.** Flowchart for combined classification and regression tree-random forest (CART-RF) method-based multivariate time-series classification.

### 3.4. Combinatorial Spatial-Temporal-Logical Filtering of Anomalous Changes

In the classification map obtained by CART-RF multivariate time-series classification, most of the misclassifications occur in the regions that correspond to scattered, fragmented, or mixed pixels. As the CART-RF method is a pixel-based classifier, the occurrence of "salt-and-pepper" noise is inevitable. To address this issue, the anomalous pixel changes were filtered after land use classification. To this end, we constructed a combinatorial spatial-temporal-logical filter for detecting most anomalous changes; this filter performed anomaly detection and correction in a 3 × 3 moving window.

The combinatorial spatial-temporal-logical filtering process consisted of single-phase spatial filtering, the temporal filtering of each pixel, and logical filtering. The single-phase spatial filter reduces the occurrence of the "salt-and-pepper" noise, while the temporal filter corrects most anomalous changes and part of the misclassified pixels. Finally, the logical filter corrects for illogical land use changes. The steps involved in the combined filtering process are as follows:

(1) Single-phase spatial filtering: spatial filtering is performed in a 3 × 3 neighbor window of a single time-phase. If the central point has been misclassified or contains no data (i.e., "salt-and-pepper" noise exists), the type of the central pixel is the majority type of the nine pixels (Figure 5a).

(2) Temporal filtering of each pixel: temporal filtering is performed on three time-phases for each pixel. The classified land use types of the pixel in year T1 and year T3 are inspected to determine whether they are identical. If the land use types in year T1 and year T3 are identical, the land use type in year T2 cannot be different from that in T1 or T3 (Figure 5b).

(3) Logical filtering: logical filtering is performed after the spatial and temporal filtering processes. Under certain circumstances, the conversion of built-up land into other subclasses is illogical. Therefore, logical filtering is necessary to prevent the appearance of illogical land use changes in the classification map. For example, in one instance, built-up land that is adjacent to other contiguous built-up lands (most likely to be urban land in reality) was converted into farmland (A1 in Figure 5c).

Since this change is not possible, the conversion was considered illogical. In another example, built-up land that is not adjacent to other built-up lands (most likely to be rural settlements in reality) was converted into other subclass at an earlier date (A2 in Figure 5c); if A2 was classified as built-up land at a later date, the land use change would be deemed illogical too.

Combinatorial spatial-temporal-logical filtering (i.e., the combined use of the filtering schemes shown in Figure 5a–c) was performed, in units of years, on the results of the land use classification step. The class of each pixel was jointly determined based on the classification of 27 pixels (3 × 3 spatial-3 time-phase filtering) and the results of logical filtering. Combinatorial spatial-temporal-logical filtering was performed repeatedly for the entirety of the period of study (with the exception of the first and last years).

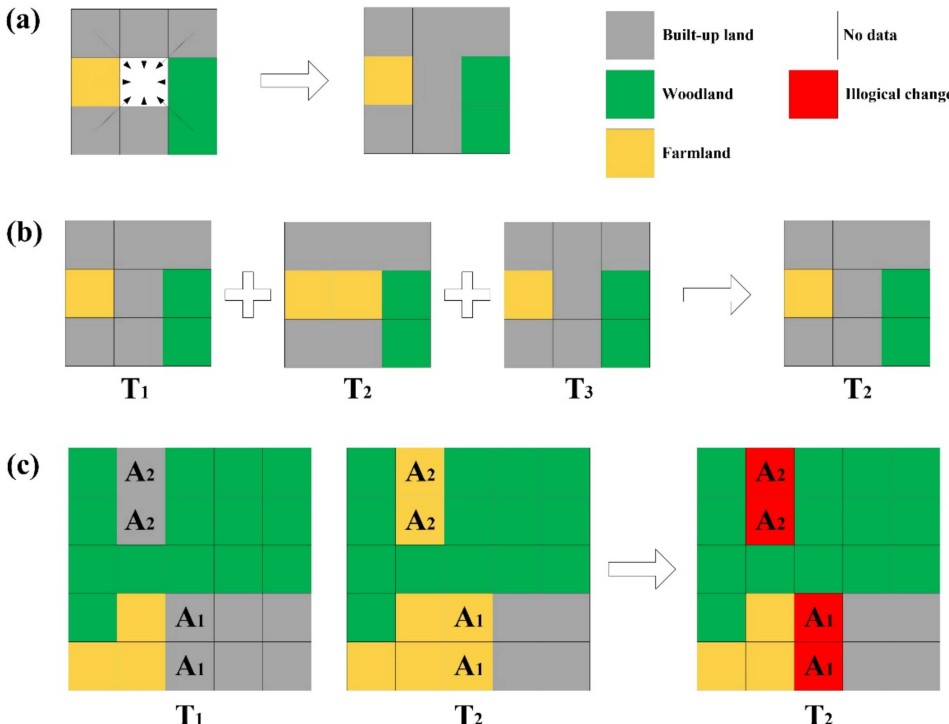

**Figure 5.** Illustration of various filters. Examples of (**a**) spatial filtering, (**b**) temporal filtering, and (**c**) logical filtering.

## 4. Results and Analysis

### 4.1. Analysis of Land Use Classification Results

#### 4.1.1. Accuracy of Land Use Classification

The accuracy of the land use classification results was validated using land use survey data and Google Earth images [43]. The classifications were first validated via comparisons with land use survey data. Next, validating classifications were performed using 2015 Google Earth images, in order to check the accuracy of the classifications in the area of study, during the later years of the period of study. In the former case, the classification results for the area of study were compared with the land survey data whereas in the latter case, validation was performed using 4147 sample pixels that were (approximately) uniformly distributed across the area of study.

To analyze the overall accuracy of each remote sensing image classifier, the results obtained using the different classifiers were compared with each other and evaluated using validation data [44]. To analyze the effectiveness of the multivariate time-series, classification was performed using the NDVI time-series as well as the multivariate (NDVI, NDWI, and NDSI) time-series. The classification based

on the NDVI time-series was performed using the RF method; combinatorial spatial-temporal-logical filtering was performed in this case. The classification based on the multivariate time-series was performed using the CART-RF model; combinatorial spatial-temporal-logical filtering was performed in this case too. The results, which are listed in Table 3, show that the NDVI time-series-based land use classification had an overall accuracy of approximately 80% and a Kappa coefficient of approximately 0.78. Moreover, the classification accuracy for farmlands, woodlands, and water bodies was high, highlighting the advantages of multiphase MODIS data for the classification of remote sensing images. Nonetheless, when the NDVI data were combined with the NDWI and NDSI data to form a multivariate time-series and the CART-RF model was used to perform land use classification, the overall classification accuracy was even higher at more than 90%. This increase in the classification accuracy may be attributed to two factors: firstly, the various land use types are more readily distinguishable in a multivariate time-series than in a single-index NDVI-based time-series. Secondly, the CART-RF multivariate time-series classifier fully exploits the advantages and characteristics of each index in land use classification and thus has a higher classification accuracy.

It can be seen from Table 3 that the classification accuracies associated with farmlands, built-up lands, and water bodies were high, whereas that for grasslands was relatively lower. One possible reason for this is that accuracy assessments have a large degree of uncertainty, since they are based on a relatively small number of samples. Another possible reason for this observation is that the phenological characteristics of grasslands are similar to those of other land use types (e.g., farmlands that are in season only once a year); this could also decrease its classification accuracy. Since grasslands and farmlands (especially non-irrigated farmlands) display similar spectral and temporal (periodic) properties, it is very difficult to distinguish these two land use types with absolute reliability.

**Table 3.** Overall accuracies of classified land use and land survey data.

| Land Use Type | 2000 | | 2005 | | 2010 | | 2015 | |
|---|---|---|---|---|---|---|---|---|
| | NDVI | Multi-Index | NDVI | Multi-Index | NDVI | Multi-Index | NDVI | Multi-Index |
| Farmland | 82.29 | 90.65 | 82.36 | 90.21 | 81.19 | 90.32 | 82.43 | 90.71 |
| Built-up land | 59.60 | 87.72 | 63.30 | 88.22 | 60.21 | 88.73 | 62.58 | 88.96 |
| Water body | 84.06 | 91.70 | 83.70 | 91.58 | 85.33 | 90.83 | 85.42 | 91.64 |
| Woodland | 76.38 | 91.76 | 72.82 | 92.32 | 71.40 | 92.41 | 75.99 | 91.89 |
| Grassland | 64.88 | 83.22 | 66.67 | 81.54 | 68.35 | 81.70 | 67.29 | 80.47 |
| Kappa | 0.77 | 0.89 | 0.79 | 0.89 | 0.79 | 0.90 | 0.78 | 0.89 |
| Overall Accuracy | 79.35 | 90.83 | 81.50 | 90.65 | 81.42 | 90.87 | 81.87 | 90.60 |

NDVI represents accuracy for land use classification based only on NDVI time-series whereas Multi-Index represents accuracy for land use classifications based on multivariate NDVI, NDWI, and NDSI time-series.

## 4.1.2. Land Use Classification Results

There are many lakes in Jiangsu, such as Lake Tai and Hongze Lake, to name a few. In addition, the province is also one of the most important grain-producing areas of China. In general, farmland is the primary land use type in Jiangsu, followed by built-up land, and then water bodies. In 2015, farmland, built-up land, and water bodies accounted for 56.8%, 22.9%, and 15.6%, respectively of the total land area in Jiangsu. The proportion of land use type in Jiangsu province over the 2000–2015 period is shown in Figure 6. The results show that the combinatorial spatial-temporal-logical filtering method was effective in reducing the occurrence of "salt-and-pepper" noise and even eliminated most of the anomalous pixel changes, thus increasing the accuracy of land use classification. However, the combined filtering method also eliminated a few of the fragmented classification pixels. While this is conducive for highlighting the land use trends in the area of study, some of the finer details may have also been obscured by the filter.

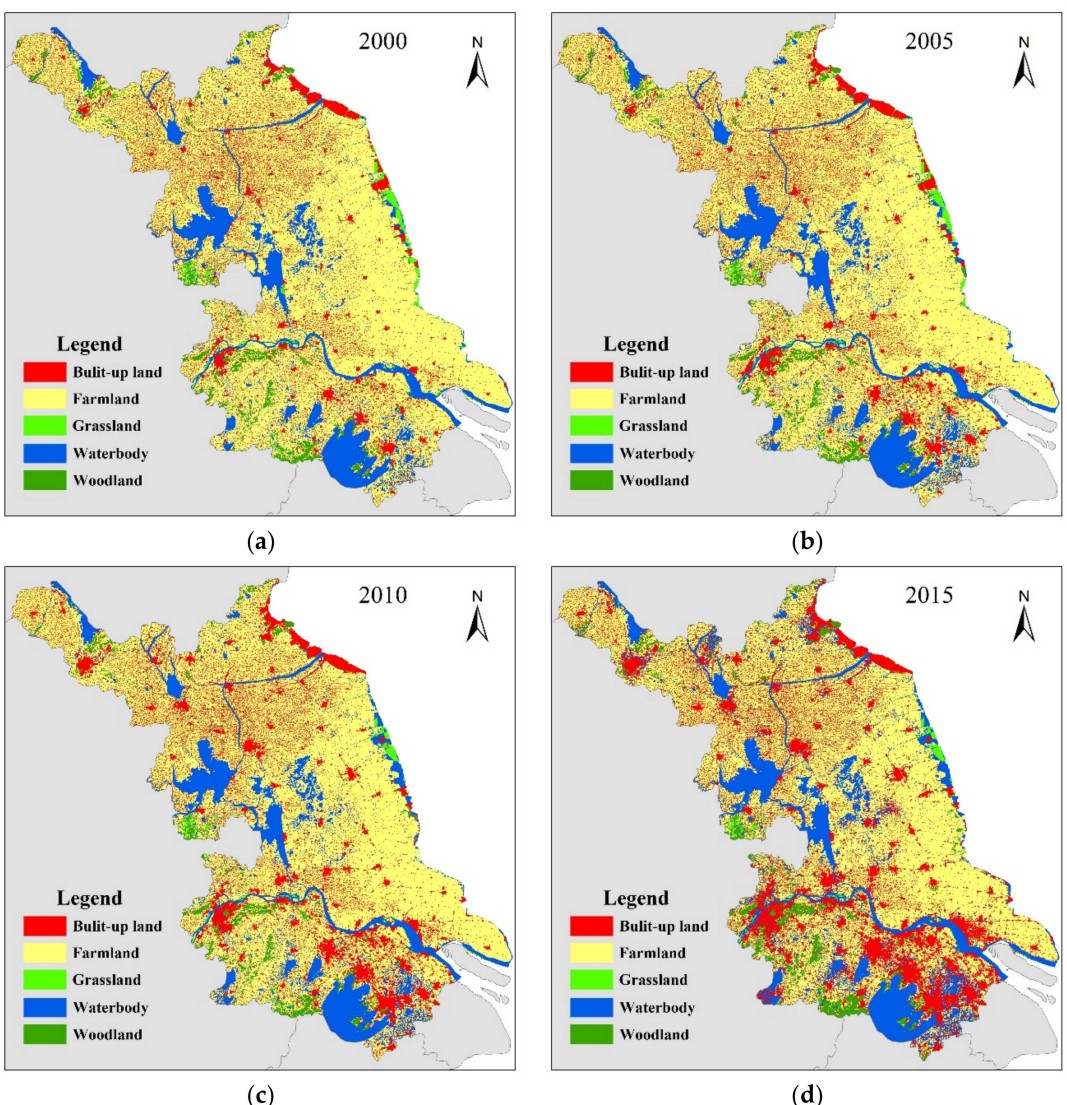

**Figure 6.** Land use maps of Jiangsu for 2000–2015.

*4.2. Spatiotemporal Analysis of Land Use Changes*

4.2.1. Overall Trend in Land Use Changes in Jiangsu Province for 2000–2015

The rapid economic development and urbanization of Jiangsu province has been accompanied by significant changes in its land use over the 2000–2015 period, and large tracts of farmland have been converted into built-up land. The proportion of farmland decreased from 68.1% in 2000 to 56.8% in 2015, while the proportion of built-up land increased from 14.3% in 2000 to 22.9% in 2015 (Table 4).

**Table 4.** Land use statistics for Jiangsu for 2000–2015.

| Year | Percentage of Study Area (%) | | | | |
|------|----------|---------------|------------|----------|-----------|
| | Farmland | Built-Up Land | Water Body | Woodland | Grassland |
| 2000 | 68.1 | 14.3 | 13.0 | 3.3 | 1.3 |
| 2005 | 66.7 | 15.3 | 13.6 | 3.4 | 1.1 |
| 2010 | 63.5 | 18.5 | 14.0 | 3.1 | 0.9 |
| 2015 | 56.8 | 22.9 | 15.6 | 4.0 | 0.8 |

Table 5 illustrates how the primary land use types of Jiangsu province changed from 2000 to 2015. It can be seen that the greatest change in Jiangsu's land use occurred in the case of farmland, which was reduced by 13,717.1 km$^2$. This was followed by built-up land, whose area increased by 9813.5 km$^2$. Finally, the area of water bodies increased by 3609.5 km$^2$. Woodlands and grasslands occupy relatively small proportions of the area of study. Compared with primary land use types (farmland, built-up land, and water body), they did not change obviously. In addition, due to their small proportions of the study area, they were more affected by the classification uncertainty. Therefore, the changes in these land use types were excluded from the analysis in this paper.

**Table 5.** Transition matrix of land use in Jiangsu for 2000 to 2015 (km$^2$).

| 2000–2015 | Farmland | Built-Up Land | Water Body | Woodland | Grassland | Change |
|---|---|---|---|---|---|---|
| Farmland | 55,762.6 | 9835.0 | 3353.4 | 1005.7 | 79.0 | −13,717.1 |
| Built-up land | 173.5 | 14,196.3 | 293.5 | 15.4 | 31.6 | 9813.5 |
| Water body | 81.2 | 249.9 | 13,038.5 | 4.9 | 8.6 | 3609.5 |
| Woodland | 160.3 | 122.6 | 23. 7 | 3087.6 | 1.9 | 748.6 |
| Grassland | 141.0 | 120.0 | 307.2 | 7.4 | 708.7 | −454.5 |

Figure 7 illustrates the changes in built-up lands, farmlands, and water bodies that occurred in Jiangsu province over the 2000–2015 period. It is evident that the built-up lands had undergone massive growth, whereas the farmlands had shrunk significantly; the areas where the land use changes had occurred also showed a significant degree of aggregation. The area covered by water bodies had increased slightly, and the distribution of the water bodies over the area of study was highly fragmented.

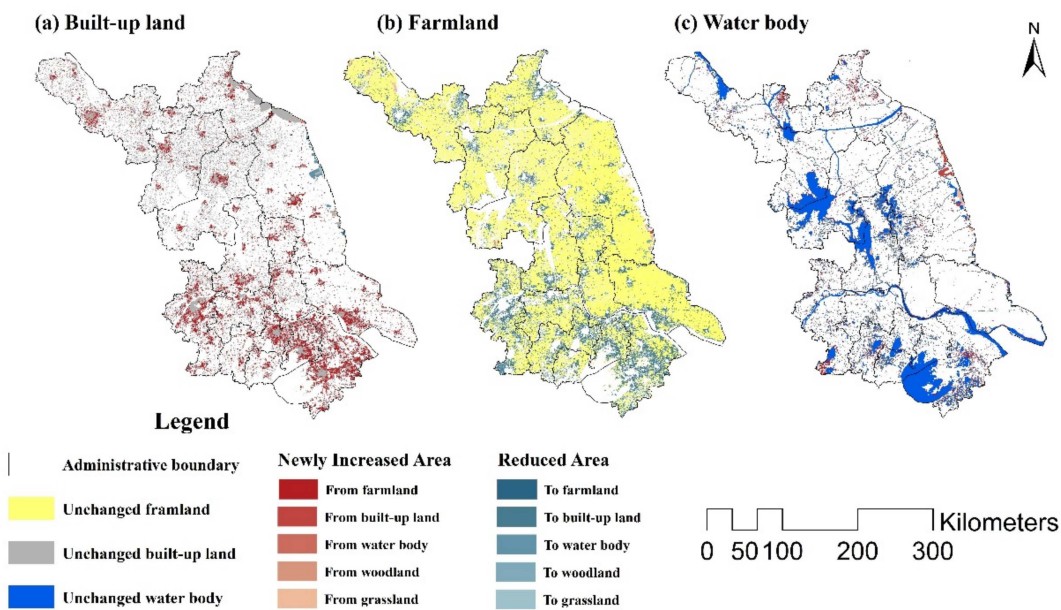

**Figure 7.** Maps of changes in land use in Jiangsu between 2000 and 2015.

### 4.2.2. Temporal Variations in Land Use

The period of study was divided into three periods, to facilitate an analysis of the yearly rate of land use change in Jiangsu province for each year. It was found that the rate of land use change was relatively low in the 2000–2005 period but then increased somewhat in the 2005–2010 period, before reaching its maximum in the 2010–2015 period. In each period, the primary changes in the land use type were decreases in farmland and expansions in built-up land. The other changes in the land use types were relatively small in comparison (Table 6).

**Table 6.** Land use change rates during three periods (km$^2$/year).

| Year | Farmland | Built-Up Land | Water Body | Woodland | Grassland |
|------|----------|---------------|------------|----------|-----------|
| 2000–2005 | −300.0 | 198.5 | 125.0 | 13.6 | −37.1 |
| 2005–2010 | −644.6 | 655.9 | 85.6 | −61.9 | −34.9 |
| 2010–2015 | −1798.8 | 1108.2 | 516.1 | 193.3 | −18.8 |

The decrease in farmland from 2000 to 2015 is consistent with the overall rate of change in the land use; the rate of decrease in farmland increased over time, reaching its maximum of 1798.8 km$^2$/year between 2010 and 2015 (Table 6). The large-scale decrease in farmland was caused by two factors: (1) the acceleration of economic development and urbanization in Jiangsu, which led to the conversion of farmlands into built-up lands on a massive scale and (2) increased environmental awareness, which led the Jiangsu government to actively promote the restoration of farmland into woodlands and lakes; large tracts of farmland were thus converted into woodlands and water bodies, which further decreased the area occupied by farmlands(Figure 8).

From Figure 9 and Table 6, it can be seen that the expansion of built-up land occurred at the expense of farmland and that this trend became more pronounced over time. The rate of expansion of built-up land in the 2000–2005 period was 198.5 km$^2$/year and increased to 655.9 km$^2$/year during 2005–2010. The highest rate of expansion of built-up land, which was 1108.2 km$^2$/year, occurred during 2010–2015, and was 1.7 times that for the 2005–2010 period. This rapid expansion of built-up land was driven by accelerated urbanization, the rapid expansion of urban and rural infrastructure, and China's urgent need for to develop secondary and tertiary industries.

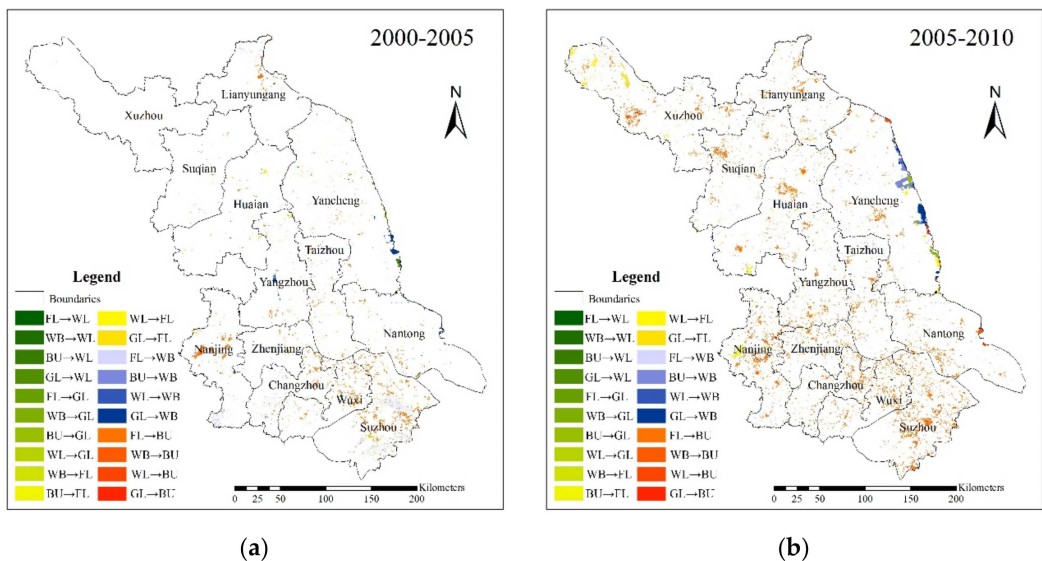

(**a**)                                              (**b**)

**Figure 8.** *Cont.*

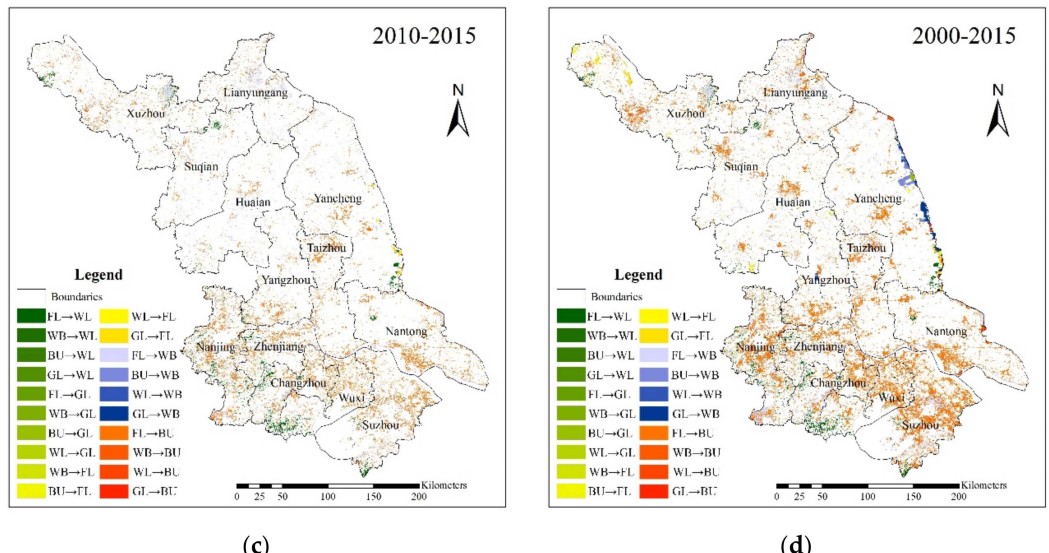

**Figure 8.** Jiangsu land use change maps for different periods. **Note:** FL, Farmland; BU, Built-up land; WB, Water Body; WL, Woodland; GL, Grassland.

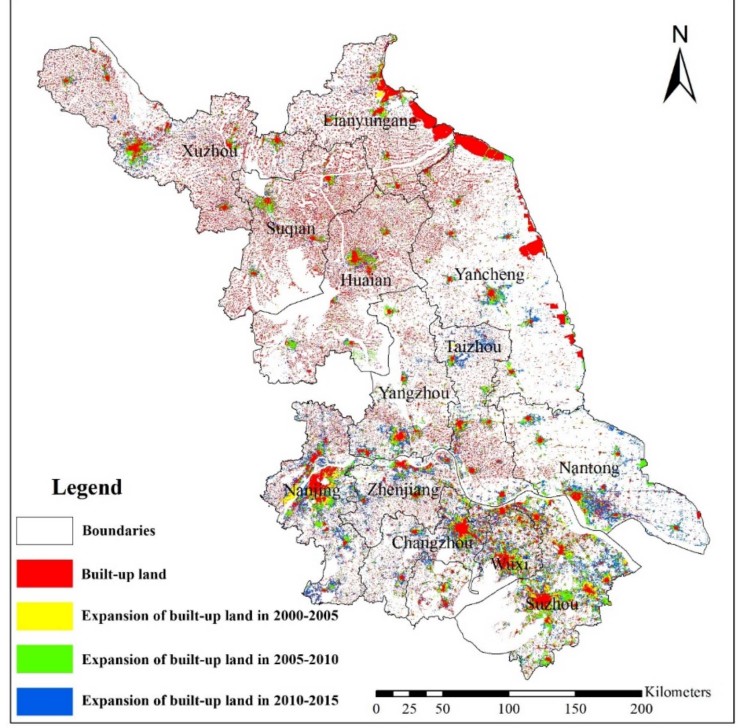

**Figure 9.** Expansion of construction areas in Jiangsu for 2000–2015.

### 4.2.3. Spatial Heterogeneity of Land Use Change

Although significant changes occurred in land use throughout Jiangsu province over the 2000–2015 period, the changes varied from region to region. During our analysis, we found that the decrease in farmland and increase in built-up land showed signs of aggregation. Therefore, we performed a detailed analysis of the land use changes in Jiangsu from a spatial perspective as well.

Table 7 shows that decreases in farmland and increases in built-up land had occurred throughout the area of study. However, these changes were not uniform with respect to the different divisions of Jiangsu province. The decreases in farmland and increases in built-up land were the most pronounced in

southern Jiangsu (Nanjing, Wuxi, Changzhou, and Suzhou) and the banks of Yangtze River (Nantong), whereas land use in northern Jiangsu (Xuzhou, Lianyungang, Huai'an, Yancheng, and Suqian) did not change significantly over the same period. The trend in the land use changes in central Jiangsu lay somewhere between those for southern and northern Jiangsu. With the exception of a few regions such as Xuzhou and Huai'an, the area covered by woodland generally increased throughout Jiangsu. Moreover, apart from Xuzhou, the area occupied by water bodies also increased throughout Jiangsu. These trends in land use change were caused by two factors. Firstly, the rapid economic development of Jiangsu aided the conversion of farmland into built-up land. Secondly, woodlands and lakes were restored gradually throughout Jiangsu province; this was driven by increased environmental awareness and the implementation of government policies, including those mandating the restoration of farmland into woodlands and lakes.

**Table 7.** Net changes in land use ($km^2$).

|  | Built-Up Land | Farmland | Water | Woodland | Grassland |
|---|---|---|---|---|---|
| Nanjing | 1198.4 | −1777.3 | 390.6 | 188.3 | 0.0 |
| Wuxi | 882.0 | −1089.3 | 89.1 | 120.4 | −2.2 |
| Changzhou | 728.7 | −1094.5 | 209.4 | 161.8 | −5.4 |
| Suzhou | 1841.3 | −2364.7 | 456.8 | 70.6 | −3.9 |
| Zhenjiang | 544.5 | −707.2 | 104.9 | 59.7 | −2.0 |
| Nantong | 1036.5 | −1110.6 | 121.1 | 18.1 | −65.0 |
| Yangzhou | 532.4 | −733.2 | 210.6 | 20.0 | −29.7 |
| Taizhou | 614.4 | −794.8 | 176.3 | 4.4 | −0.3 |
| Xuzhou | 784.0 | −1085.7 | 356.0 | −20.8 | −33.5 |
| Lianyungang | 443.9 | −857.5 | 410.4 | 4.3 | −1.2 |
| Huai'an | 405.0 | −607.5 | 209.1 | −16.2 | 9.7 |
| Yancheng | 401.2 | −816.0 | 678.0 | 64.7 | −327.9 |
| Suqian | 401.0 | −678.7 | 220.9 | 49.9 | 6.9 |

The rapid development of secondary and tertiary industries in Jiangsu has reduced the role of agriculture in its economy. This has also promoted the conversion of farmland into built-up land. Although the trends in land use change were generally the same throughout Jiangsu, there were significant differences between northern and southern Jiangsu in terms of the scale and rate of their land use changes. The scale and rate of land use changes, that is, the decreases in farmland and increases in built-up land, were the greatest in southern Jiangsu and on the banks of Yangtze River (Figure 10). The land use changes in northern Jiangsu occurred over a much smaller area in comparison, and the rate of land use change was also relatively lower in this region.

This spatial heterogeneity can mainly be attributed to economic, location, and policy factors. Southern Jiangsu and the banks of Yangtze River have high levels of economic development and large populations. Moreover, the influence of Shanghai, the economic center of China, is also much stronger in these regions; these factors led to massive land use changes. Furthermore, southern Jiangsu officially implemented the "Southern Jiangsu Modernization Demonstration Zone Masterplan", which promotes the large-scale conversion of farmland into built-up land, in 2013. The changes in the water areas mainly occurred around the coastal cities of Jiangsu province. In 2009, China proposed a new development strategy for Jiangsu's coastal regions; large tracts of farmland and woodland around coastal cities such Lianyungang and Yancheng were thus converted into aquaculture farms. This significantly increased the coverage of water areas.

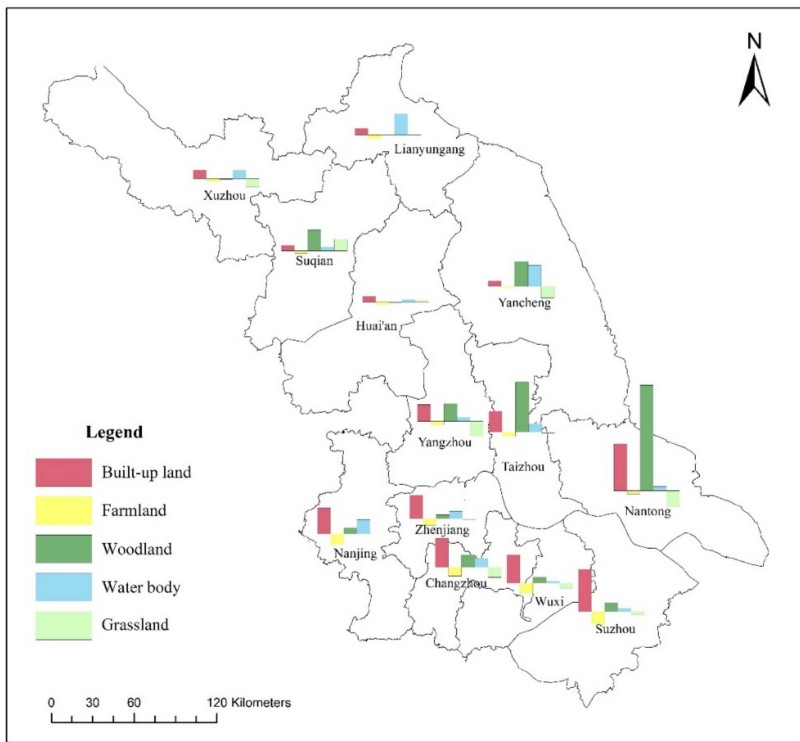

**Figure 10.** Land use change dynamics in Jiangsu province.

## 5. Discussion

### 5.1. The Advantages of the CART-RF Method

The experiments showed that NDVI time series did not perform well for classifying built-up land and grassland. Therefore, we proposed CART-RF method based on multivariate time-series. The proposed CART-RF method can effectively make full use of the multivariate time-series, so the accuracy of classification results has been greatly improved.

We determined the averaged curves with one standard deviation for NDVI for all five land use types. This was done for 20 reference areas corresponding to each land use type. The averaged NDVI curves show typical season-related characteristics, with the minimum value being observed in the winter and the maximum value in the summer (Figure 11). Although the averaged NDVI curves were smoothed, they did exhibit sporadic, small-scale fluctuations, predominantly in the spring. These small fluctuations primarily occurred in farmlands and grasslands and were most likely caused by the weather, such as the plum rain season, in Jiangsu province. However, fluctuations this small would have a limited effect on the NDVI curves. Thus, the phenological characteristics of the NDVI can be employed for land use classification.

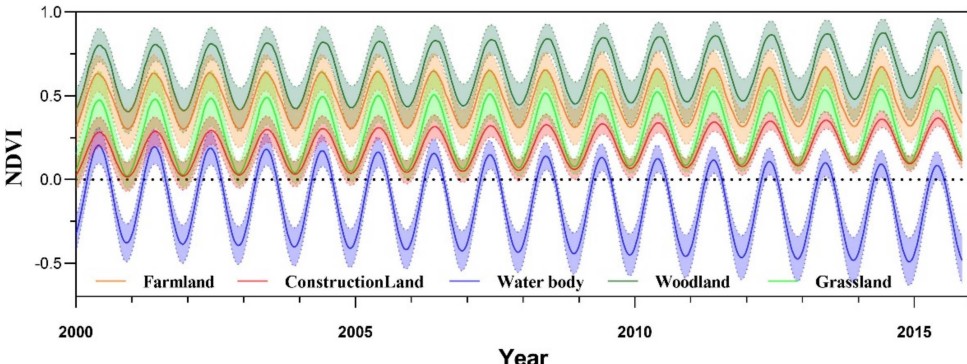

**Figure 11.** Seasonal mean NDVI curves with one standard deviation for all five land use types for reference region of Jiangsu province.

However, the potential for misclassifications was higher in the case of some land use types, owing to there being significant overlap between their NDVI curves. Generally, the NDVI values for farmland and woodland were 0.3–0.8, and their NDVI characteristics are very similar. Moreover, with the exception of some differences during winter, the NDVI values for the other seasons were nearly identical. Meanwhile, the built-up land and grassland also showed very similar NDVI characteristics. In addition, there was no significant difference between the seasons, with the exception of summer, when the NDVI value for grassland is higher than that for built-up land. We analyzed the reference dataset and found this was because of the low resolution of the MODIS dataset. Built-up land often includes urban green fields, because of which it shows NDVI characteristics similar to those of grassland. The NDVI curve for water bodies was markedly lower than those for the other land use types and thus could be distinguished readily. The average NDVI of water bodies in the summer is often higher than zero, and these values look abnormal. On the other hand, our analysis of the reference dataset did not reveal any problems regarding the location of the reference areas. Therefore, assuming that these changes in the NDVI were attributable to natural processes, it can be concluded that they were caused by the high biomass content around rivers and shallow lakes.

Compared to the range of changes in the averaged NDVI curves, those in the averaged NDWI curves were smaller (Figure 12). Although the NDVI curves for farmland and woodland were very similar, the range of changes in the NDWI curve for farmland was larger than that for woodland, with farmland exhibiting a more well-defined curve with distinct seasonal peaks. However, there was significant overlap between the curves for farmland and woodland. Similarly, built-up land and grassland showed very similar phenological characteristics as evidenced by their NDVI curves. However, their NDWI curves were different to the point where they could be distinguished readily. The NDVI curve of the water bodies was distinctly different from those of the other land use types and the same was true for the NDWI curves as well.

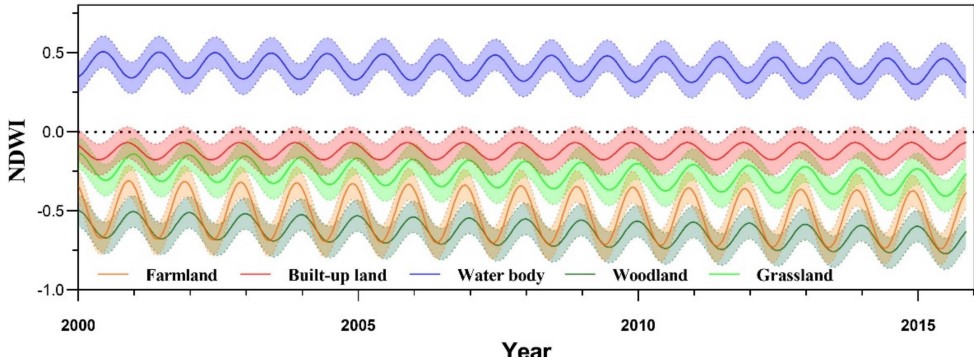

**Figure 12.** Seasonal mean NDWI curves with one standard deviation for all five land use types for reference region of Jiangsu province.

Finally, the overall range of seasonal changes in the NDSI curves was the smallest (Figure 13). The phenological characteristics of farmland, built-up land, water bodies, and grassland were very similar, as evidenced by their NDSI curves. The change range and minimum value of the NDSI curve for woodland could be seen readily, making its curve easy to distinguish from those of the other land use types. Further, while the NDVI curves for farmland and woodland were very similar, their NDWI curves were not but still exhibited significant overlap, which affected the classification results. On the other hand, the NDSI curves of woodland and farmland showed some degree of overlapping. However, their ranges were very different. Thus, woodland and farmland could be distinguished with ease.

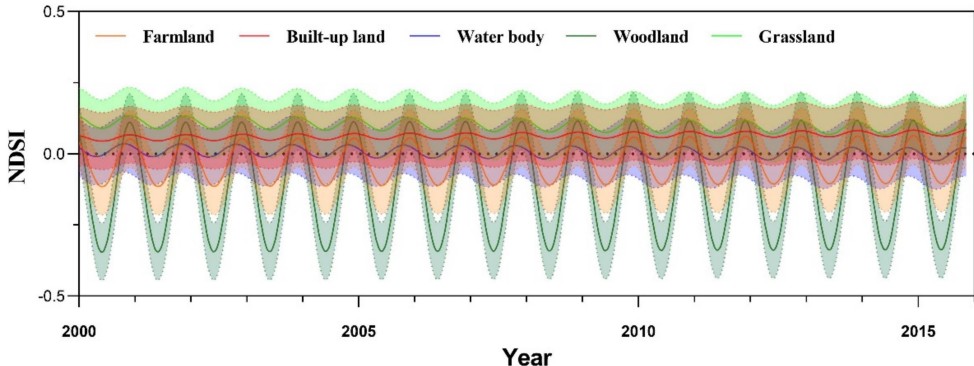

**Figure 13.** Seasonal mean NDSI curves with one standard deviation for all five land use types for reference region of Jiangsu province.

One can effectively distinguish water bodies based only on the NDVI time-series. However, the probability of mislabeling for farmland and woodland as well as that for built-up land and grassland is high. By using both NDVI and NDWI time-series, built-up land and grassland can be classified efficiently. However, the chances of the misclassification of farmland and woodland still exist. Therefore, in addition to the NDVI and NDWI time-series, we also used the NDSI time-series to further distinguish between farmland and woodland.

The spatial resolution of MODIS data is low, and a few mixed pixels are inevitable [45]. The mixed pixels affect the classification accuracy [46]. The highest accuracy of land use classification is woodland (nearly 92%), the lowest classification accuracy was grassland (approximately 81%), and thus there is uncertainty in the classification results. Our results can show the long-term change trend of major land use types, but cannot show small changes in Jiangsu. Statistically, we did not analyze the changes of woodland and grassland in detail, because insignificant changes may lead to wrong results and affect our correct analysis of regional land use changes.

In the future, we will consider other TSRS data with higher spatial resolution (such as those from the LandSat, and Sentinel satellites) to test the classification accuracy of the proposed method. In addition, it can be seen that each land use type exhibited different characteristics with respect to the different remote sensing indices. Therefore, it is necessary to use multivariate TSRS data for land use classification. In this study, we built multivariate time-series using NDVI, NDWI, and NDSI data. However, multivariate time-series data may also be built using other remote sensing indices, such as EVI (Enhanced Vegetation Index), and NDBI (Normalized Difference Built-up Index), to name a few. In future works, we will compare and analyze other remote sensing indices and try to determine the optimal combination of remote sensing indices for land use classification, in order to improve the land use classification accuracy over large areas.

### 5.2. Land Use Change in Yangtze River Delta

Our research and related studies have shown that the land use in the Yangtze River Delta (YRD) and Jiangsu has changed obviously [47]. In 2000–2015, YRD presented the obvious change of land use,

mainly showing: farmland, grassland steady decline; and built-up land, woodland and water bodies continue to rise; unused land remaining the same. Farmland is one of the most common land use types from which built-up land and woodland changed. Consistent with the YRD, Jiangsu has been undergoing the continuous expansion of built-up land which occupied lots of farmland. The difference between Jiangsu and the YRD is that (1) the process of conversion from farmland to built-up land is different. In the YRD, the conversion speed was relatively slow from 2000 to 2005, then the fastest from 2005 to 2010, and decreased from 2010 to 2015. In Jiangsu, the speed was slow from 2000 to 2005, then increased from 2005 to 2010, and became faster from 2010 to 2015. (2) The YRD's water bodies had increased from 2000 to 2015, but obviously decreased from 2005 to 2010. While water bodies in Jiangsu increased slightly. (3) The YRD's woodland increased slightly. However, in Jiangsu, the woodland first increased from 2000 to 2005, then decreased slightly from 2005 to 2010, and increased substantially from 2010 to 2015.

Previous studies prove that natural factors, social economic factors, and land use policy are the major driving factors of land use. Although, similar driving factors can be found in relevant literature, but specific factors vary in different regions. We used Partial Least Squares Regression method to analyze the driving force of land use change in Jiangsu, and found the main driving force: urbanization process, the rapid development of economy, and the rapid increase of the population. This would explain the reason that the YRD's built-up land expansion from 2005 to 2010 reached the fastest speed, and Jiangsu's did from 2010 to 2015. The land use policy also affected land use change. In general, the Chinese government put forward policies to promote the development of the industrial economy and urbanization. It increased the demand for built-up land (Tables 4 and 5). Since 2005, the Jiangsu's government has strengthened the cooperation of northern and southern Jiangsu. One of the most important methods was industry transformation from south Jiangsu to northern Jiangsu, then in 2005–2010 built-up land was obviously increased in north Jiangsu.

The rapid expansion of built-up land may lead to some problems [48], such as the loss of farmland [49] and ecological degradation [50]. To address the issues, the Chinese government has adopted a series of policies and measures, including the 'Grain to Green' project, 'Returning Cropland to Lake and Wetlands' project, etc. Existing research has pointed out that almost 153,694 km$^2$ of farmland was converted into woodland in the whole Yangtze River Delta due to the implementation of the 'Grain to Green' project [51]. Jiangsu has also carried out this plan, which accounted for the increase of woodland from 2010 to 2015. To protect the ecological and environment, the Chinese government implemented the 'Returning Cropland to Lake and Wetlands' project. Under the influence of this project, the lakes in southern Jiangsu increased. In addition, Jiangsu also carried out coastal ecological restoration and protection policies, which increased water bodies in northern Jiangsu coastal areas. To maintain enough farmland for food security, the Chinese government was obliged to carry out farmland development, consolidation, and reclamation to compensate the loss of farmland. As a result, 151,349 km$^2$ of woodland, 40,128 km$^2$ of grassland, and 16,009 km$^2$ of water bodies were converted to farmland. Jiangsu is also actively promoting farmland protection policies. Although, the farmland in Jiangsu was decreasing during the study period, but the farmland in some areas of northern Jiangsu was increasing. In 2018, farmland protection became an obliged responsibility of the municipal government and is assessed by a higher level government. Since then, the loss of farmland has gradually slowed down.

## 6. Conclusions

Based on the MOD13A1 and MODIS09A1 datasets, a multivariate NDVI, NDWI, and NDSI time-series was constructed using an asymmetric Gaussian model. A CART-RF multivariate time-series classifier and combinatorial spatial-temporal-logical filtering model were also constructed. Using these methods, we constructed a dataset of land use in Jiangsu province from 2000 to 2015 and analyzed the spatiotemporal changes in the land use in Jiangsu during this period. The primary conclusions of this study are as follows:

(1) The results indicated that the CART-RF classifier can be used successfully to classify MODIS multivariate time-series into land use data, with the overall classification accuracy being greater than 90%. Therefore, the approach is suitable for analyzing the changes in land use. By comparing the classification accuracies for the NDVI time-series alone and the multivariate time-series, it was found that an increase in the number of classification features (variables) improves the classification accuracy. For example, the overall classification accuracies of the NDVI-based land use classification process were 79.35%, 81.50%, and 81.42% for the 2000, 2005, and 2010 datasets, respectively. However, the addition of the NDWI and NDSI indices increased the overall classification accuracies to 90.83%, 90.65%, and 90.87%, respectively. Thus, the thorough mining of the multispectral and multiphase features from MODIS data is essential for improving land use classification based on MODIS data.

(2) The proposed combinatorial spatial-temporal-logical method filtering method was effective in removing the individual pixels with anomalous changes. This, in turn, improved the accuracy of land use classification. We constructed a combinatorial spatial-temporal-logical filtering method to reduce the effect of "salt-and-pepper" noise and hence remove the individual pixels with anomalous changes. As MODIS data has relatively low spatial resolution, these data contain a large number of mixed pixels, which can result in misclassifications by the CART-RF model. The use of the combinatorial spatial-temporal-logical filter eliminated most of these misclassifications, thus improving the overall classification accuracy.

(3) There were significant spatiotemporal heterogeneities in the land use changes in Jiangsu province over the 2000–2015 period. The total area with land use changes was large, approximately 9400 km$^2$, which accounts for 8.8% of Jiangsu's area. Decreases in farmland and increases in built-up land were significant throughout the province. Moreover, the rate of change of land use in Jiangsu province was not uniform over time and in fact had accelerated. It was relatively low during the 2000–2005 period, slightly higher for 2005–2010, and the highest during 2010–2015. From a spatial perspective, there were significant differences between southern and northern Jiangsu in terms of the scale of land use changes in each region: the decrease in farmland and increase in built-up land mostly occurred on the banks of Yangtze River and southern Jiangsu regions such as Wuxi, Suzhou, Nanjing, and Nantong. However, land use in Northern Jiangsu did not change significantly, in contrast. The differences between northern and southern Jiangsu in the terms of their land use change trends were mainly caused by economic, location, and policy factors. To begin with, southern Jiangsu and the banks of Yangtze River had higher levels of economic development, larger populations, and are more strongly influenced by Shanghai; all these factors contributed to the acceleration of land use changes in these regions. Secondly, the implementation of policies such as the "Southern Jiangsu Modernization Demonstration Zone Masterplan" also significantly affected land use changes in Jiangsu.

**Author Contributions:** L.Q. designed and performed the experiments, wrote the draft; Z.C. designed the research, and revised the manuscript; M.L. funded and supervised research, and revised the manuscript. All authors reviewed the manuscript.

**Funding:** This research was funded by the National Key Research and Development Program of China, grant number 2017YFB0504205; the National Natural Science Foundation of China, grant number 41571378 and the Natural Science Research Project of Higher Education in Anhui Provence, grant number KJ2017A307.

**Acknowledgments:** We thank the National Aeronautics and Space Administration and the Chinese Academy of Sciences Geospatial Data Cloud for free use of MODIS images.

**Conflicts of Interest:** There is no conflict of interest. The funding sponsors had no role in the design of the study; in the collection, analyses, or interpretation of data; in the writing of the manuscript; and in the decision to publish the results.

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
