# Peer review of "CART-RF Classification with Multifilter for Monitoring Land Use Changes Based on MODIS Time-Series Data: A Case Study from Jiangsu Province, China"

_sustainability, doi:10.3390/su11205657_

Round 1

Reviewer 1 Report

The manuscript describes an application of a classification method based on MODIS time-series data to detect the land cover change in the Jangsu province. The method seems to work as observed when comparing the k-accuracy for multyindex vs NDVI. This is a quite relevant result, since Chinese regions are very interesting for land use changes.

However, the paper is too focused on technical aspects while neglecting many relevant issues concerning land use change in a deltaic region. This reduces the potential audience of the work.

About the technical issues, I suggest to explain what is the "salt and pepper noise", as mentioned throughout the paper.

I suggest to divide the discussion section in 2 paragraph: the first dealing with the technical aspects (remote sensing methods), and the second discussing the scientific aspects (land use change).

Deltaic regions are subject to many natural and human-related drivers and it would be interesting to compare the findings with other areas (both Chinese and non-Chinese). Check for example the following papers:

Xu, et al (2018). Ecosystem services trade-offs and determinants in China's Yangtze River Economic Belt from 2000 to 2015. Science of the Total Environment, 634, 1601-1614.   Gaglio et al (2017). Land use change effects on ecosystem services of river deltas and coastal wetlands: case study in Volano–Mesola–Goro in Po river delta (Italy). Wetlands ecology and management, 25(1), 67-86.   Chen et al (2018). Spatiotemporal dynamics of coastal wetlands and reclamation in the Yangtze Estuary during past 50 years (1960s–2015). Chinese geographical science, 28(3), 386-399. Gaglio et al (2019). Ecosystem services approach for sustainable governance in a brackish water lagoon used for aquaculture. Journal of Environmental Planning and Management, 1-24.   Islam et al (2015). Implications of agricultural land use change to ecosystem services in the Ganges delta. Journal of environmental management, 161, 443-452.   Moreover, the China is subject to an extensive program for ecological restoration (the Grain-for-Green project) whose effects on land use changes are well documented in literature. Therefore, the author should discuss the observed land use change with respect to these other cases. Did the project failed in the Jiangsu province? Why?     

Reviewer 2 Report

The study title: "CART-RF Classification with Multifilter for Monitoring Land Use Changes Based on MODIS Time-Series Data: A Case Study From Jiangsu Province, China" in my opinion is excellent, a very well executed study. The methods are explained in detail, necessary in these types of studies and also suitable. The results and the discussions are clear and interesting. Throughout the study the questions asked at the beginning have been answered; the CART-RF classifier can be used to classify MOdis multivariate time-series into land use data, and the results are good for took a well management of the area. 

The figure 10 for example seems fantastic to me, visually the soil changes by province are perceived. Also the great cartographic work in the study. 

Suggestions:

in the study area: there is talk about Jiangsu is one of the most economically developed proveinces in China, ... please can your put a comparative example, number or position in the provinces in China, to locate the reader of the magnitute of those who are talking about.

Table 2- in the column Source, leave the space or decrease the font one point.

Tables, in general, leave only one decimal or none, the same in the text.

Data sources pag.4 line 124, scale 1:10000 ? meters, milles? km?

In all the text 1530,73km2, remove the decimals

pag 7 line 193, 3     3?

pag. 8 line 209-211 it's necessary this paragraph?

Figure 6- increase the font size of the legend

Pag 13. line 350-351- it's your decision, althought o don't share it, because I think that surface has not changed in comparison to the others, ...

Round 2

Reviewer 1 Report

The authors addressed all the comments. The technical part is well documented and interesting, while the discussion of the scientific aspects of the land use change may help the paper to capture a broader audience. I think the paper is suitable for publication in the journal.

There is an error in references list: missing number for the reference Gaglio et al. 2019 Wetlands Ecol Manage 2017, 25, 67–86 (lines 729-731)

This manuscript is a resubmission of an earlier submission. The following is a list of the peer review reports and author responses from that submission.